## Reduced-Cost Construction of Jacobian Matrices for High-Resolution Inversions of Satellite Observations of Atmospheric Composition

Hannah Nesser<sup>1</sup>, Daniel J. Jacob<sup>1</sup>, Joannes D. Maasakkers<sup>2</sup>, Tia R. Scarpelli<sup>3</sup>, Melissa P. Sulprizio<sup>1</sup>, 5 Yuzhong Zhang<sup>4,5</sup>, Chris H. Rycroft<sup>1</sup>

<sup>1</sup>School of Engineering and Applied Sciences, Harvard University, Cambridge, MA, USA
 <sup>2</sup>SRON Netherlands Institute for Space Research, Utrecht, the Netherlands
 <sup>3</sup>Department of Earth and Planetary Sciences, Harvard University, Cambridge, MA, USA
 <sup>4</sup>Key Laboratory of Coastal Environment and Resources of Zhejiang Province, School of Engineering, Westlake University, Hangzhou, Zhejiang, China

Institute of Advanced Technology, Westlake Institute for Advanced Study, Hangzhou, Zhejiang, China

### Correspondence to: Hannah Nesser (hnesser@g.harvard.edu)

- Abstract. Global high-resolution observations of atmospheric composition from satellites can greatly improve our understanding of surface emissions through inverse analyses. Variational inverse methods can optimize surface emissions at any resolution but do not readily quantify the error and information content of the posterior solution. The information content of satellite data may be <u>much</u> lower than its coverage would suggest because of failed retrievals, instrument noise, and error correlations that propagate through the inversion. Analytical solution <u>of</u> the inverse problem provides closed-form.
- characterization of posterior error statistics and information content but requires the construction of the Jacobian matrix that relates emissions to atmospheric concentrations. Building the Jacobian matrix is computationally expensive at high resolution because it involves perturbing each emission element, typically individual grid cells, in the atmospheric transport model used as forward model for the inversion. We propose and analyze two methods, reduced-dimension and reduced-rank, to construct the Jacobian matrix at greatly decreased computational cost while retaining information content. Both methods
- are two-step iterative procedures that begin from an initial native-resolution estimate of the Jacobian matrix constructed at no computational cost by assuming that atmospheric concentrations are most sensitive to local emissions. The reduced-dimension method uses this estimate to construct a Jacobian matrix on a multiscale grid that maintains high resolution in areas with high information content and aggregates grid cells elsewhere. The reduced-rank method constructs the Jacobian matrix at native resolution by perturbing the leading patterns of information content given by the initial estimate. We
- demonstrate both methods in an analytical Bayesian inversion of GOSAT methane satellite data with augmented information content over North America in July 2009. We show that both methods reproduce the results of the native-resolution inversion while achieving a factor of <u>four</u> improvement in computational performance. The reduced-dimension method

| Deleted: In fact, t          |  |
|------------------------------|--|
| Deleted: orders of magnitude |  |
| Deleted: s                   |  |
| Deleted: to                  |  |

Deleted: 4School of Engineering, Westlake University, Hangzhou, Zhejiang Province, Chin

Formatted: Superscript

Deleted: 4

produces an exact solution at lower spatial resolution while the reduced-rank method solves the inversion at native resolution in areas of high information content and defaults to the prior estimate elsewhere.

### **1** Introduction

Satellite observations of atmospheric composition provide a powerful resource to improve our knowledge of emissions (Streets et al., 2013). However, the inverse analyses used to infer emissions from observed atmospheric concentrations are subject to large errors from the measurements and <u>from</u> the inversion <u>procedure</u>. Conducting inverse analyses of satellite
data to quantify emissions at high resolution is of considerable interest but may be limited by data quality in ways that are difficult to quantify and that may compromise the results. Here we present two methods to conduct high-resolution inversions of satellite observations that optimize the information content of the observations while providing full error statistics and minimizing computational cost.

- Inverse analyses infer emissions by fitting the observed atmospheric concentrations to a chemical transport model (CTM) that simulates atmospheric concentrations as a function of emissions (Brasseur and Jacob, 2017). The CTM represents the forward model for the inverse problem. The solution is generally obtained in a Bayesian framework by minimizing a cost function regularized by a prior emissions estimate. The optimal (posterior) emissions estimate corresponds to the minimum of the cost function. This minimum is typically found using a numerical (variational) method, often employing the adjoint of
- the CTM to compute the cost function gradient (e.g., Daescu et al., 2000; Elbern and Schmidt, 2001; <u>Quélo et al., 2005</u>; Henze et al., 2007). However, the numerical solution provides no explicit characterization of the solution's error or information content. Methods of estimating the error exist (e.g., Chevallier et al., 2007; Meirink et al., 2008; Koohkan et al., 2013), but these approaches are computationally expensive, incomplete, and rarely applied in practice.
- In the common case where the observed atmospheric concentrations depend linearly on emissions and the error statistics can be assumed to be normally or log-normally distributed, the Bayesian optimization problem has an analytical solution including closed-form expressions for the posterior emissions estimate, its error statistics, and its information content (Rodgers, 2000; Maasakkers et al., 2019). The analytical solution requires explicit construction of the Jacobian matrix of the forward model, K = ∂y/∂x ∈ ℝ<sup>m×n</sup>, which represents the sensitivity of the simulated concentrations y ∈ ℝ<sup>m</sup> to the 65 emission state vector x ∈ ℝ<sup>n</sup> (Brasseur and Jacob, 2017). The elements of y are individual observations and the elements of x are the emissions optimized by the inversion, often grid cells in a two-dimensional emissions field. When m ≫ n, as for inversions of satellite observations, the Jacobian can be constructed column-wise by conducting n + 1 CTM simulations to
- perturb each of the state vector elements  $x_i$  and obtain the corresponding column  $\partial \mathbf{y} / \partial x_i$ . Even on massively parallel computing clusters, the computational cost of conducting these simulations can limit the size of the state vector  $\mathbf{x}$  and
- therefore the resolution at which inversions are conducted (Turner and Jacob, 2015). However, once the Jacobian matrix is

Deleted: itself

Deleted: : Ouélo et al., 2005

Deleted: Bayesian

**Deleted:** Chevallier et al., 2007; Koohkan et al., 2013; Meirink et al., 2008)

Deleted: optimized

Deleted: T

constructed, inversions can be conducted at essentially no additional computational cost, allowing study of the solution's sensitivity to changes in the specification of inversion parameters, <u>error statistics</u>, prior assumptions, and the number and type of observations.

An illustrative example is the inversion of satellite observations to infer methane emissions. Methane is an important greenhouse gas but the spatial and temporal distribution of emissions is highly uncertain (Saunois et al., 2019). Satellite observations of atmospheric methane columns can inform emission estimates (Jacob et al., 2016). This was first shown with

- data from the SCIAMACHY satellite instrument (2003 2012) with nadir pixel resolution of 30 x 60 km<sup>2</sup> (Bergamaschi et al., 2009, 2013; Houweling et al., 2014; Wecht et al., 2014). More recent inversions used observations from the TANSO-FTS instrument aboard the GOSAT satellite (2009 present) with 10-km diameter pixels approximately 250 km apart along-and cross-track (Monteil et al., 2013; Alexe et al., 2015; Turner et al., 2015; Maasakkers et al., 2019). The Tropospheric Monitoring Instrument (TROPOMI) aboard the Sentinel-5 precursor satellite, launched in October 2017, now provides daily,
- global retrievals of atmospheric methane columns at 5.5 x 7 km<sup>2</sup> nadir pixel resolution, increasing coverage by orders of magnitude relative to GOSAT (Veefkind et al., 2012). However, TROPOMI's methane retrieval has only a ~3% success rate for daytime scenes limited by dark surfaces (water), clouds, high aerosol loadings, and variable surface albedo and topography, resulting in heterogeneously distributed observations (Hu et al., 2018; Hasekamp et al., 2019). Inversions of TROPOMI data must attempt to capture the high resolution and density of observations where appropriate while recognizing
- the limitations in information content resulting from data sparsity and observational errors.

80

Several methods have been proposed to decrease the computational cost of high-resolution analytical inversions by optimally reducing the dimension or rank of the <u>observations or</u> state vector. Approaches that decrease the dimension of the <u>observation vector (e.g., Xu, 2007) reduce the computational cost of solving the inversion but not of constructing the lacobian matrix. Approaches that decrease the dimension of the state vector lower the cost of both computations. Reduced-dimension methods, solve inversions on a multiscale emission grid of dimension *k* < *n* for which the construction of the Jacobian matrix K ∈ R<sup>m×k</sup> is computationally tractable. Bocquet et al. (2011) and Bocquet and Wu (2011), defined a method to <u>select a multiscale grid from a limited array of allowable grids that preserve resolution where the observations have the highest information content. Turner and Jacob (2015) used prior emissions information to group together similar grid cells
using a Gaussian mixture model, but the criteria used to define similarity were subjective and did not consider the information content of the forward model or the observations. Other approaches that decreased the dimension of the state vector assumed knowledge of the Jacobian matrix (e.g., Rigby et al., 2011; Thompson and Stohl, 2014; Ray et al., 2015;
</u></u>

|                        | Deleted: ,                                              |
|------------------------|---------------------------------------------------------|
|                        |                                                         |
|                        | Deleted: ; Hu et al., 2018                              |
|                        |                                                         |
|                        |                                                         |
|                        |                                                         |
|                        |                                                         |
|                        |                                                         |
|                        |                                                         |
|                        |                                                         |
|                        | Deleted: (Bocquet et al., 2011; Turner and Jacob, 2015) |
|                        | Deleted: Bocquet et al. (2011)                          |
| ~                      | Deleted: find                                           |
| $\langle \neg \rangle$ | Deleted: the optimal                                    |
| $( \land )$            | Deleted: n                                              |

Deleted: all

Deleted: including

Deleted: , but this requires a large computational investment

| Deleted: ; Lunt et al., 2016; Ray et al., 2015; Rigby et al., 2011; |  |
|---------------------------------------------------------------------|--|
| Thompson and Stohl, 2014)                                           |  |
| Deleted: (Bousserez and Henze, 2018; Spantini et al., 2015)         |  |

3

Lunt et al., 2016; Liu et al., 2017). Reduced-rank methods generate an approximation of the posterior solution at the original dimension *n* by solving the inversion in the directions of highest information content. The reduced-rank method proposed by

<sup>110</sup> Spantini et al. (2015) assumed knowledge of the Jacobian matrix. Bousserez and Henze (2018) and Miller et al. (2020)

avoided explicit construction of the Jacobian matrix by estimating the directions of highest information content, but their approach is effective only if a small number of directions explain most of the information content.

Here we present two methods to construct the Jacobian matrix for a native *n*-dimensional state vector and maximize the information content of the inverse analysis using k < n forward model simulations. The reduced-dimension method generates

- a multiscale grid that preserves native resolution where information content is highest and aggregates grid cells elsewhere. The resulting reduced-dimension Jacobian matrix  $\mathbf{K}_{RD} \in \mathbb{R}^{m \times k}$  solves the inversion exactly on the multiscale grid. The reduced-rank method constructs a Jacobian matrix  $\mathbf{K}_{\Pi} \in \mathbb{R}^{m \times n}$  along the dominant patterns of information content in the system, allowing the approximation of the inverse solution at native resolution. In both cases, a low-cost initial estimate of the Jacobian matrix is updated using k forward model simulations where k is selected by the user based on the information
- content of the observing system and the available computational resources. We demonstrate both methods in a 1-month inversion of satellite data.

### 2 Methods

This section presents the reduced-dimension and reduced-rank methods of <u>constructing the</u> Jacobian matrix, <u>Following a</u> review of the standard analytical inverse framework (Section 2.1), we describe optimal reductions in both dimension and

140 rank for an inverse system with a known native-resolution Jacobian matrix K ∈ R<sup>m×n</sup> (Section 2.2). We then present a twostep approach to approximate an initially unknown Jacobian matrix using reductions in dimension and rank (Sections 2.3 through 2.5). For the purposes of illustration, we take the state vector to be a gridded field of <u>static</u> emissions, <u>but the</u> methods apply to <u>temporally variable emissions and more generally to any state vector.</u>

### 2.1 Analytical solution to the inverse problem

The optimal estimate **x** of a state vector **x** given a prior estimate **x**<sub>A</sub>, observation vector **y**, and normal error statistics given by prior and observational error covariance matrices **S**<sub>A</sub> and **S**<sub>O</sub>, respectively, is obtained by the minimization of the Bayesian scalar cost function  $\sqrt{J}(\mathbf{x})_{\mathbf{x}}$  (Brasseur and Jacob, 2017):

$$\mathcal{J}_{\mathbf{X}}(\mathbf{x}) = (\mathbf{x} - \mathbf{x}_{A})^{\mathrm{T}} \mathbf{S}_{A}^{-1} (\mathbf{x} - \mathbf{x}_{A}) + (\mathbf{y} - \mathbf{F}_{(\mathbf{x})})^{\mathrm{T}} \mathbf{S}_{0}^{-1} (\mathbf{y} - \mathbf{F}_{(\mathbf{x})}).$$

Here F(x) represents the forward model that simulates the observations y given x. In our application, the forward model is a CTM. The observational error covariance matrix S<sub>0</sub> includes errors from both the measurement and the forward model, which collectively form the observing system. If the forward model is linear so that F(x) = Kx + c, where  $K = \frac{\partial y}{\partial x}$  is the

Deleted: that

(1)

Deleted: boxes

Deleted: although
Deleted: any

| ( | Deleted: J   |
|---|--------------|
| ( | Deleted: (x) |
| ( | Deleted: /   |

Jacobian matrix calculated by finite difference (see Introduction) and  $\mathbf{c}$  is a constant, then an analytical solution to the cost function minimum exists that yields both the posterior estimate  $\mathbf{x}$  and its error covariance matrix  $\mathbf{S}$ :

$$\mathbf{x} = \mathbf{x}_{A} + \mathbf{S}_{A}\mathbf{K}^{T}(\mathbf{K}\mathbf{S}_{A}\mathbf{K}^{T} + \mathbf{S}_{0})^{-1}(\mathbf{y} - \mathbf{K}\mathbf{x}_{A}) = \mathbf{x}_{A} + \mathbf{S}\mathbf{K}^{T}\mathbf{S}_{0}^{-1}(\mathbf{y} - \mathbf{K}\mathbf{x}_{A}),$$
(2)  
$$\mathbf{S} = (\mathbf{K}^{T}\mathbf{S}_{0}^{-1}\mathbf{K} + \mathbf{S}_{A}^{-1})^{-1}.$$
(3)

Comparison of **S** and **S**<sub>A</sub> defines the information content of the observing system, quantified by the averaging kernel matrix  $\mathbf{A} = \partial \mathbf{x} / \partial \mathbf{x}$  that represents the sensitivity of the posterior emissions estimate  $\mathbf{x}$  to the true state  $\mathbf{x}$ . A can be calculated as  $\mathbf{A} = \mathbf{I} - \mathbf{SS}_{A}^{-1}$  or equivalently as

$$\mathbf{A} = \mathbf{S}_{\mathrm{A}} \mathbf{K}^{\mathrm{T}} (\mathbf{K} \mathbf{S}_{\mathrm{A}} \mathbf{K}^{\mathrm{T}} + \mathbf{S}_{\mathrm{O}})^{-1} \mathbf{K}.$$
<sup>(4)</sup>

Equation (4) expresses the dependence of the averaging kernel matrix on the forward model and both error covariance matrices. The diagonal elements of A are commonly referred to as the averaging kernel sensitivities. They are highest in highly observed locations with uncertain, high emissions and lowest in poorly observed areas or in regions known to have low emissions. The sum of the sensitivities, or the trace of A, measures the number of pieces of information that can be independently quantified by the observing system, known as the degrees of freedom for signal or DOFS (Rodgers, 2000).

### 2.2 Optimal reductions in dimension and rank of inverse systems

- We first consider the problem of optimally reducing the dimension and rank of an inverse system as described in Section 2.1 with a known Jacobian matrix K ∈ ℝ<sup>m×n</sup>. Figure 1 illustrates dimension and rank reductions for an emission grid over North America. The top left panel represents the original *n*-dimensional state space, i.e., the native-resolution grid. A linear transformation Γ ∈ ℝ<sup>k×n</sup> reduces the dimension of the state space from *n* to *k*. This transformation may reduce dimension discretely, as in the case of grid cell aggregation (top right panel), or non-discretely, in which case the *k* state vector components are themselves *n*-dimensional vectors (bottom right panel). A second linear transformation Γ ∈ ℝ<sup>n×k</sup> restores
- the dimension of the state space from k back to the original n. The resulting space, depicted in the bottom left, is a low-rank approximation of the original state space. The matrix  $\Pi = \Gamma^* \Gamma$  transforms the original state space to the low-rank subspace. The inverse problem can be solved in any of these four spaces, although the eigenvector corrections generated in the non-discrete reduced-dimension space (bottom right panel) would be difficult to interpret.
- We wish to define matrices  $\Gamma$  and  $\Gamma^*$  that minimize the information loss associated with reducing the dimension or rank of the state vector. Bousserez and Henze (2018) show that the projection  $\Pi$  that maximizes the probability of restoring the original full dimension state vector  $\mathbf{x}$  given the reduced dimension state vector  $\Gamma \mathbf{x}$  is given by  $\Pi = S_A^{1/2} U U^T S_A^{-1/2}$  where

 $\mathbf{U} = \mathbf{S}_{A}^{1/2} \Gamma (\Gamma \mathbf{S}_{A} \Gamma^{T})^{-1/2}.$  For a projection of this form, they show that information loss is minimized by maximizing DOFS<sub>II</sub> = Tr( $\mathbf{A}_{\Pi}$ ) = Tr( $\mathbf{U}^{T} \mathbf{S}_{A}^{-1/2} \mathbf{A} \mathbf{S}_{A}^{1/2} \mathbf{U}$ ) where  $\mathbf{A}_{\Pi}$  and  $\mathbf{A}$  are the reduced-rank and native-resolution averaging kernel matrices, respectively. Defining

$$\mathbf{Q} = \mathbf{S}_{\mathbf{A}}^{-1/2} \mathbf{A} \mathbf{S}_{\mathbf{A}}^{1/2} = \mathbf{W} \mathbf{\Sigma} \mathbf{W}^{\mathrm{T}},\tag{5}$$

Deleted:

where the columns of **W** are the eigenvectors of **Q** and **\Sigma** is a diagonal matrix of the corresponding eigenvalues ranked in 200 descending order, Bousserez and Henze (2018) show that  $\text{Tr}_{(\mathbf{A}_{\Pi})}$  is maximized for a rank k subspace when  $\mathbf{U} = \mathbf{W}_k$  where  $\mathbf{W}_k$  is the matrix of the first k columns of **W**. The corresponding optimal projection is then

 Figure 1. Dimension and rank reductions of a gridded emissions field. The linear transformation matrix Γ reduces the dimension
 of the original state space (upper left) either discretely by aggregating grid cells to generate a multiscale grid (upper right) or nondiscretely by projecting along the patterns given by the rows of Γ (lower right, with positive values in red and negative in blue). The reverse transformation Γ\* restores the dimension but not the rank, producing a low-rank subspace of the original state space (lower right). The projection Π = Γ\*Γ reduces rank but not dimension.

$$\Pi = \mathbf{S}_{A}^{1/2} \mathbf{W}_{k} \mathbf{W}_{k}^{\mathrm{T}} \mathbf{S}_{A}^{-1/2}.$$
(6)

This projection applies a dimension-reducing transformation  $\Gamma$  followed by a dimension-restoring transformation  $\Gamma^*$ :

$$\Gamma = \mathbf{W}_{k}^{T} \mathbf{S}_{k}^{-1/2},$$
(7)  

$$\Gamma^{*} = \mathbf{S}_{k}^{1/2} \mathbf{W}_{k}.$$
(8)

The columns of  $\Gamma^*$  give an eigenvector basis for the averaging kernel matrix while the diagonal of  $\Sigma$  give its eigenvalues, together defining the dominant patterns of information content. The fraction of information content explained by the first *i* columns of  $\Gamma^*$  is the sum of the *i* largest eigenvalues divided by the total DOFS (Bousserez and Henze, 2018). The eigenvalues can also be related to other measures of information content, including the Shannon and relative entropy differences (Rodgers, 2000; Xu, 2007). We will refer to the ordered list of the eigenvalues as the information content spectrum. On the basis of this spectrum, we can select *k* so that most of the information content is explained by the first *k* eigenvectors. Alternatively, we can select *k* so that all eigenvectors have a sufficiently large signal-to-noise ratio. The signalto-noise ratio SNR of the *i* th eigenvector is given by the *i* th singular value of the pre-whitened Jacobian matrix  $\mathbf{K} =$  $\Sigma_0^{-1/2} \mathbf{KS}_A^{1/2}$  and is calculated as

### where $\sigma_i$ is the *i*th ordered eigenvalue of **Q** (Rodgers, 2000),

### 230 2.3 Approximating the Jacobian matrix

Section 2.2 described optimal reductions in dimension and rank of a state vector assuming knowledge of the nativeresolution Jacobian matrix **K**. However, the n + 1 forward model simulations needed to construct **K** may be prohibitively expensive. Here we present a two-step approach to construct a reduced-dimension or reduced-rank Jacobian matrix at much lower computational cost. We start from a low-cost, native-resolution estimate  $\mathbf{K}^{(0)}$  (see below) and calculate the 235 corresponding averaging kernel matrix  $\mathbf{A}^{(0)}$ . In the reduced-dimension method, we use  $\mathbf{A}^{(0)}$  to construct a multiscale grid that maintains resolution in the areas of highest information content (top right panel of Figure 1). We generate the updated, reduced-dimension Jacobian matrix  $\mathbf{K}_{RD}^{(1)} \in \mathbb{R}^{m \times k}$  on the resulting grid using the forward model. In the reduced-rank method, we construct  $\mathbf{K}_{\Pi}^{(1)} \in \mathbb{R}^{m \times n}$  on the basis of the *k* dominant eigenvectors of  $\mathbf{A}^{(0)}$  by perturbing those patterns in the forward

 $SNR_i = \begin{bmatrix} \sigma_i \\ 1 - \sigma_i \end{bmatrix},$ 

7

### Deleted: of

# Deleted: k

**Deleted:** rate at which the information content accumulates as the number of eigenvectors increasesordered list of the eigenvalues

| Deleted: . The diagonal matri |  | veletea: |  |  | i ne | diagona | i mai | п. |
|-------------------------------|--|----------|--|--|------|---------|-------|----|
|-------------------------------|--|----------|--|--|------|---------|-------|----|

| (         | Deleted: | ۸            |
|-----------|----------|--------------|
| $ \land $ | Deleted: | Σ            |
| (         | Deleted: | <b>ΙΣ</b> -1 |

(9)

Formatted: Font: Not Bold

**Deleted:** gives the singular values of the pre-whitened Jacobian matrix  $\mathbf{K} = \mathbf{S}_0^{-1/2} \mathbf{K} \mathbf{S}_A^{1/2}$  and represents the signal-to-noise ratio of each eigenvector (Rodgers, 2000)...

model, generating an approximation of the Jacobian matrix in a reduced-rank state space (bottom left panel of Figure 1). In both methods, the updated Jacobian matrix improves the estimate of the averaging kernel matrix and its eigenvectors by incorporating information content from the forward model. We use either  $K_{RD}^{(1)}$  or  $K_{\Pi}^{(1)}$  to conduct a second update and construct the final Jacobian matrix.

In our demonstration case, we generate an initial estimate of the native-resolution Jacobian matrix  $\mathbf{K}^{(0)}$  at no cost by assuming that a local perturbation of methane emissions  $\Delta x$  [kg m<sup>-2</sup> s<sup>-1</sup>] produces local dry column mixing ratio enhancements  $\Delta y$  [mol mol<sup>-1</sup>] as determined by a simple column mass balance dependent on local wind speed and parameterized turbulent diffusion. We construct  $\mathbf{K}^{(0)}$  column-wise by assuming that observation *i* responds to emissions in 260

$$\Delta y_i = \alpha_{ij} \frac{M_{\rm air}}{M_{\rm CH_4}} \frac{Lg}{Up} \Delta x_j \tag{10}$$

### so that the elements $k_{ij}^{(0)}$ of $\mathbf{K}^{(0)}$ are given by 265

grid cell j as

$$k_{ij}^{(0)} = \frac{\partial y_i}{\partial x_j} = \alpha_{ij} \frac{M_{\text{air}}}{M_{\text{CH}_4}} \frac{Lg}{Up},\tag{11}$$

where  $\alpha_{ij} \in [0, 1]$  is a dimensionless coefficient providing a crude parameterization of turbulent diffusion,  $M_{air}$  and  $M_{CH_4}$ 270 are the molecular weights of dry air and methane, respectively, L is a ventilation length scale taken as the square root of the grid cell area, g is gravitational acceleration, U is the local wind speed taken here as 5 km h<sup>-1</sup>, and p is the surface pressure. We assume  $a_{ij} = 0.4$  for observations in grid cell j and distribute the remaining mass over the three concentric rings surrounding that cell with  $\alpha_{ii} = 0.3/8$ , 0.2/16, and 0.1/24 from the inner to outer ring. Including a representation of turbulent diffusion increases the spatial coverage of the dominant patterns of information content; the exact form of the 275 parameterization (e.g., the number of rings used or the values of  $\alpha_{ii}$ ) is unimportant.

The reduced-dimension and reduced-rank methods rely on characterizing the dominant patterns of information content of the observing system using the initial estimate of the averaging kernel matrix  $\mathbf{A}^{(0)}$  corresponding to  $\mathbf{K}^{(0)}$ .  $\mathbf{A}^{(0)}$  can provide a good approximation of A even if the initial estimate of the Jacobian matrix  $\mathbf{K}^{(0)}$  is crude because the averaging kernel matrix depends strongly on the specified prior and observational error covariance matrices  $S_A$  and  $S_O(\underline{Eq.}(4))$  and because, by assuming that observed concentrations are most sensitive to local emissions,  $\mathbf{K}^{(0)}$  generates the highest information content where the observations are densest. This information content structure can then be refined by a two-step update.

### Deleted: and therefore

Deleted: row

Deleted: representation

Deleted: This

| Formatted: Font: Not Bold |  |
|---------------------------|--|
| Formatted: Font: Not Bold |  |
| Deleted: equation         |  |

### 2.4 Constructing the reduced-dimension Jacobian matrix

- In an inverse system with a known native-resolution Jacobian matrix **K**, a reduced-dimension Jacobian matrix **K**<sub>RD</sub> can be constructed on a multiscale grid that maintains native resolution where information content is highest and <u>aggregates grid</u> cells elsewhere (top right panel of Figure 1). We refer to the state vector elements of this multiscale grid as clusters. An optimal multiscale grid maximizes the total DOFS and the averaging kernel sensitivities of each state vector element, referred to here as the DOFS per cluster. To construct this grid, we first define the state vector as a single element encompassing the inversion domain. We then add the native-resolution grid cells with the highest averaging kernel
- sensitivities to the state vector one-by-one, removing them from the original state vector element. For each new element  $x_i$ , we calculate the corresponding Jacobian matrix column  $\partial \mathbf{y} / \partial x_i$  and the resulting increase in DOFS. When the DOFS stabilize, we add instead clusters of two or more native-resolution grid cells and repeat this procedure. Clusters can be generated by, for example, K-means clustering, which aggregates spatially proximate grid cells. We repeat this process, increasing cluster size, until all native-resolution grid cells are allocated to the multiscale grid and the corresponding
- reduced-dimension Jacobian matrix  $\mathbf{K}_{RD}$  is constructed. The DOFS convergence criteria and the sequence of cluster sizes can be selected to achieve the desired state vector dimension.

We apply this approach beginning with our initial estimate  $\mathbf{K}^{(0)}$  (Section 2.3) in a two-step update that iteratively improves the multiscale grid. Algorithm 1 describes this process in detail. Briefly, the information content for the initial multiscale

- grid is given by  $\mathbf{A}^{(0)}$ , which <u>we use to identify the grid cells with the highest information content and construct a multiscale</u> grid <u>as described above. We compute the corresponding reduced-dimension Jacobian matrix  $\mathbf{K}_{RD}^{(1)}$ , introducing information content from the forward model. We identify the state vector elements where the forward model contributes the most information content by comparing the sensitivities given by the updated reduced-dimension averaging kernel matrix  $\mathbf{A}_{RD}^{(1)}$  to the sensitivities given by  $\mathbf{A}^{(0)}$ . We disaggregate the clusters with the largest differences and update the reduced-dimension</u>
- Jacobian, generating  $\mathbf{K}_{RD}^{(2)}$ . Convergence is rapid and we find no need for further iteration. The analytical inversion can then be solved exactly on the multiscale grid using  $\mathbf{K}_{RD}^{(2)}$ .

### Algorithm 1: Reduced-dimension Jacobian matrix construction

Given a native-resolution state vector with dimension  $n_{\star}$  a state vector encompassing the entire domain with dimension  $n_{RD} = 1_{\star}$  and  $A^{(0)}_{\Lambda}$ 315 and  $A^{(0)}_{RD}$  the  $n \times n$  and  $1 \times 1$  initial estimates of the averaging kernel matrix, respectively, and an initial cluster size of one nativeresolution grid cell:

<u>1: Add the *J* clusters with the highest diagonal values of  $A^{(0)}$  to the state vector and update its dimension  $n_{RD} = n_{RD} + J_z$ </u>

2: Perturb in the forward model those *J* clusters and the background cluster to generate the  $m \times n_{\text{RD}}$  reduced-dimension Jacobian matrix  $\mathbf{K}_{\text{RD}}^{(1)}$  and the corresponding  $n_{\text{RD}} \times n_{\text{RD}}$  averaging kernel matrix  $\mathbf{A}_{\text{RD}}^{(1)}$ .

|  | De | leted: | clusters |
|--|----|--------|----------|
|--|----|--------|----------|

| Deleted: T                                                    |
|---------------------------------------------------------------|
| Deleted: identifies                                           |
| Deleted: sensitivities                                        |
| Deleted: even given the crude estimate of the Jacobian matrix |
| Deleted: . We then                                            |
| Deleted: and                                                  |
|                                                               |

**Deleted:** The information content associated with both  $K_{RD}^{(1)}$  and  $K_{RD}^{(2)}$  includes contributions from prior emissions estimates, the observations, and the forward model. As a result, cC

Formatted: Font: Not Bold

3: If the difference in DOFS per cluster ΔDPC = Tr(A<sup>(k)</sup><sub>D</sub>)/n<sub>kD</sub> - Tr(A<sup>(k)</sup><sub>RD</sub>)/n<sub>kD</sub> - Cr(A<sup>(k)</sup><sub>RD</sub>)/n<sub>kD</sub> - Cr(A<sup>(k)</sup><sub>RD</sub>)/n<sub>kD</sub>/n<sub>kD</sub>/n<sub>kD</sub>/n<sub>kD</sub>/n<sub>kD</sub>/n<sub>kD</sub>/n<sub>kD</sub>/n<sub>kD</sub>/n<sub>kD</sub>/n<sub>kD</sub>/n<sub>kD</sub>/n<sub>kD</sub>/n<sub>kD</sub>/n<sub>kD</sub>/n<sub>kD</sub>/n<sub>kD</sub>/n<sub>kD</sub>/n<sub>kD</sub>/n<sub>kD</sub>/n<sub>kD</sub>/n<sub>kD</sub>/n<sub>kD</sub>/n<sub>kD</sub>/n<sub>kD</sub>/n<sub>kD</sub>/n<sub>kD</sub>/n<sub>kD</sub>/n<sub>kD</sub>/n<sub>kD</sub>/n<sub>kD</sub>/n<sub>kD</sub>/n<sub>kD</sub>/n<sub>kD</sub>/n<sub>kD</sub>/n<sub>kD</sub>/n<sub>kD</sub>/n<sub>kD</sub>/n<sub>kD</sub>/n<sub>kD</sub>/n<sub>kD</sub>/n<sub>kD</sub>/n<sub>kD</sub>/n<sub>kD</sub>/n<sub>kD</sub>/n<sub>kD</sub>/n<sub>kD</sub>/n<sub>kD</sub>/n<sub>kD</sub>/n<sub>kD</sub>/n<sub>kD</sub>/n<sub>kD</sub>/n<sub>kD</sub>/n<sub>kD</sub>/n<sub>kD</sub>/n<sub>kD</sub>/n<sub>kD</sub>/n<sub>kD</sub>/n<sub>kD</sub>/n<sub>kD</sub>/n<sub>kD</sub>/n<sub>kD</sub>/n<sub>kD</sub>/n<sub>kD</sub>/n<sub>kD</sub>/n<sub>kD</sub>/n<sub>kD</sub>/n<sub>kD</sub>/n<sub>kD</sub>/n<sub>kD</sub>/n<sub>kD</sub>/n<sub>kD</sub>/n<sub>kD</sub>/n<sub>kD</sub>/n<sub>kD</sub>/n<sub>kD</sub>/n<sub>kD</sub>/n<sub>kD</sub>/n<sub>kD</sub>/n<sub>kD</sub>/n<sub>kD</sub>/n<sub>kD</sub>/n<sub>kD</sub>/n<sub>kD</sub>/n<sub>kD</sub>/n<sub>kD</sub>/n<sub>kD</sub>/n<sub>kD</sub>/n<sub>kD</sub>/n<sub>kD</sub>/n<sub>kD</sub>/n<sub>kD</sub>/n<sub>kD</sub>/n<sub>kD</sub>/n<sub>kD</sub>/n<sub>kD</sub>/n<sub>kD</sub>/n<sub>kD</sub>/n<sub>kD</sub>/n<sub>kD</sub>/n<sub>kD</sub>/n<sub>kD</sub>/n<sub>kD</sub>/n<sub>kD</sub>/n<sub>kD</sub>/n<sub>kD</sub>/n<sub>kD</sub>/n<sub>kD</sub>/n<sub>kD</sub>/n<sub>kD</sub>/n<sub>kD</sub>/n<sub>kD</sub>/n<sub>kD</sub>/n<sub>kD</sub>/n<sub>kD</sub>/n<sub>kD</sub>/n<sub>kD</sub>/n<sub>kD</sub>/n<sub>kD</sub>/n<sub>kD</sub>/n<sub>kD</sub>/n<sub>kD</sub>/n<sub>kD</sub>/n<sub>kD</sub>/n<sub>kD</sub>/n<sub>kD</sub>/n<sub>kD</sub>/n<sub>kD</sub>/n<sub>kD</sub>/n<sub>kD</sub>/n<sub>kD</sub>/n<sub>kD</sub>/n<sub>kD</sub>/n<sub>kD</sub>/n<sub>kD</sub>/n<sub>kD</sub>/n<sub>kD</sub>/n<sub>kD</sub>/n<sub>kD</sub>/n<sub>kD</sub>/n<sub>kD</sub>/n<sub>kD</sub>/n<sub>kD</sub>/n<sub>kD</sub>/n<sub>kD</sub>/n<sub>kD</sub>/n<sub>kD</sub>/n<sub>kD</sub>/n<sub>kD</sub>/n<sub>kD</sub>/n<sub>kD</sub>/n<sub>kD</sub>/n<sub>kD</sub>/n<sub>kD</sub>/n<sub>kD</sub>/n<sub>kD</sub>/n<sub>kD</sub>/n<sub>kD</sub>/n<sub>kD</sub>/n<sub>kD</sub>/n<sub>kD</sub>/n<sub>kD</sub>/n<sub>kD</sub>/n<sub>kD</sub>/n<sub>kD</sub>/n<sub>kD</sub>/n<sub>kD</sub>/n<sub>kD</sub>/n<sub>kD</sub>/n<sub>kD</sub>/n<sub>kD</sub>/n<sub>kD</sub>/n<sub>kD</sub>/n<sub>kD</sub>/n<sub>kD</sub>/n<sub>kD</sub>/n<sub>kD</sub>/n<sub>kD</sub>/n<sub>kD</sub>/n<sub>kD</sub>/n<sub>kD</sub>/n<sub>kD</sub>/n<sub>kD</sub>/n<sub>kD</sub>/n<sub>kD</sub>/n<sub>kD</sub>/n<sub>kD</sub>/n<sub>kD</sub>/n<sub>kD</sub>/n<sub>kD</sub>/n<sub>kD</sub>/n<sub>kD</sub>/n<sub>kD</sub>/n<sub>kD</sub>/n<sub>kD</sub>/n<sub>kD</sub>/n<sub>kD</sub>/n<sub>kD</sub>/n<sub>kD</sub>/n<sub>kD</sub>/n<sub>kD</sub>/n<sub>kD</sub>/n<sub>kD</sub>/n<sub>kD</sub>/n<sub>kD</sub>/n<sub>kD</sub>/n<sub>kD</sub>/n<sub>kD</sub>/n<sub>kD</sub>/n<sub>kD</sub>/n<sub>kD</sub>/n<sub>kD</sub>/n<sub>kD</sub>/n<sub>kD</sub>/n<sub>kD</sub>/n<sub>kD</sub>/n<sub>kD</sub>/n<sub>k</sub>

<u>6: Perturb in the forward model the disaggregated grid cells to generate the final  $m \times n_{RD}$  reduced-dimension Jacobian matrix  $\mathbf{K}_{RD}^{(2)}$ .</u>

### 2.5 Constructing the reduced-rank Jacobian matrix

In an inverse system with a known native-resolution Jacobian matrix  $\mathbf{K}$ , a reduced-rank approximation of the Jacobian matrix  $\mathbf{K}_{\Pi}$  can be constructed by calculating the linear relationship between emissions and observations for the most important patterns of information content rather than for individual or aggregate grid cells. A low-rank Jacobian corresponds to the state space shown in the bottom left panel of Figure 1. We showed in Section 2.2 that the leading patterns of

340 to the state space shown in the bottom left panel of Figure 1. We showed in Section 2.2 that the leading patterns of information content are given by the columns of the dimension-restoring transformation  $\Gamma^*$  (Eq. (8)). For any selected value of *k*, the *k* leading patterns span a rank-*k*, dimension-*n* subspace of the original information content space. A Jacobian matrix can be constructed within this space by calculating the model response to perturbations of these patterns. The response of the forward model **F** to the *j*th normalized eigenvector  $\mathbf{y}_i^* \in \mathbb{R}^n$ , given by the *j*th column of  $\Gamma^*$ , is

$$\mathbf{y}_{j} = \frac{\mathbf{F}(\mathbf{x}_{A} + \beta \boldsymbol{\gamma}_{j}^{*}) - \mathbf{F}(\mathbf{x}_{A})}{\beta},$$
(12)

where  $\beta$  is any scalar sufficiently large to ensure numerical stability. The model responses  $\mathbf{y}_{j}$ ,  $j \in \{1, ..., k\}$  form the columns of the matrix  $\mathbf{K}_{\omega} \in \mathbb{R}^{m \times k}$ , which is the Jacobian matrix for an inverse system with a reduced-dimension state space spanned by the first *k* eigenvectors of the information content, illustrated by the bottom right panel of Figure 1. This reduceddimension Jacobian must be transformed to the original state dimension to enable physical interpretation of the posterior results. Bousserez and Henze (2018), following Bocquet et al. (2011)<sub>w</sub> show that the reduced-dimension Jacobian matrix  $\mathbf{K}_{\omega}$ is given by  $\mathbf{K}_{\omega} = \mathbf{K}\Gamma^*$  and the reduced-rank Jacobian matrix  $\mathbf{K}_{\Pi}$  by  $\mathbf{K}_{\Pi} = \mathbf{K}\Pi = \mathbf{K}\Gamma^*\Gamma$ . Thus, the reduced-rank Jacobian can be calculated from the reduced-dimension Jacobian by  $\mathbf{K}_{\Pi} = \mathbf{K}_{\omega}\Gamma$ . The resulting Jacobian has dimension  $m \times n$  and rank *k*. 355

In an inverse system without a known Jacobian matrix, the reduced-rank Jacobian matrix approximation can be constructed in a two-step update that iteratively improves the patterns of information content used as perturbations. Algorithm 2 describes this process in detail. Briefly, we use the initial estimate of the Jacobian matrix  $\mathbf{K}^{(0)}$  (Section 2.3) to calculate the corresponding averaging kernel matrix  $\mathbf{A}^{(0)}$  and the matrix of its eigenvectors  $\mathbf{\Gamma}^{*(0)}$ . When calculating  $\mathbf{\Gamma}^{*(0)}$ , we select the

 $k^{(0)}$  eigenvectors that have a signal-to-noise ratio greater than some threshold. We use the signal-to-noise criterion, which is

10

Deleted: equation

Deleted:

Deleted: W

stricter than the information content criterion, to account for the errors in the initial estimate of the information content. We 365 compute the forward model response to each of the eigenvectors using Eq. (12) and transform the resulting reduced-Deleted: equation dimension Jacobian  $\mathbf{K}_{\omega}^{(1)}$  to the full-dimension state space with  $\mathbf{K}_{\Pi}^{(1)} = \mathbf{K}_{\omega}^{(1)} \mathbf{\Gamma}^{(0)}$ . We calculate the associated averaging kernel matrix  $\mathbf{A}_{\Pi}^{(1)}$  and the matrix of its eigenvectors  $\mathbf{\Gamma}_{\Pi}^{*(1)}$ . Because  $\mathbf{K}_{\Pi}^{(1)}$  is a reduced-rank approximation, its spectrum of information content is discontinuous at  $k^{(0)}$ . We therefore use the spectrum of information content associated with the initial, full-rank estimate  $\mathbf{A}^{(0)}$  to select the rank  $k^{(1)}$  of the second update and calculate  $\mathbf{\Gamma}_{\mathbf{n}}^{*(1)}$ . We use the  $k^{(1)}$  eigenvectors that span most of the information content from the initial estimate and construct an updated reduced-rank Jacobian matrix 370 approximation  $\mathbf{K}_{\Pi}^{(2)}$  as above. The resulting Jacobian matrix  $\mathbf{K}_{\Pi}^{(2)}$  is a rank  $\approx k^{(1)}$  approximation that accurately quantifies the forward model where the observing system has high information content and loses accuracy in areas with lower information content where the observations are least able to constrain emissions. The resulting posterior solution is accurate in areas with high information content and defaults to the prior estimate elsewhere. 375 Algorithm 2: Reduced-rank Jacobian matrix construction Given a native-resolution state vector with dimension n,  $A^{(0)}$  an  $n \times n$  initial estimate of the averaging kernel matrix, and  $S_A$  the  $n \times n$ prior error covariance matrix: 1: Complete the eigendecomposition of the  $n \times n$  matrix  $\mathbf{Q}^{(0)} = \mathbf{S}_{\mathbf{A}}^{-1/2} \mathbf{A}^{(0)} \mathbf{S}_{\mathbf{A}}^{1/2} = \mathbf{W}^{(0)} \mathbf{\Sigma}^{(0)} \mathbf{W}^{(0)}^{\mathrm{T}}$ ; <u>2: Select k so that 1</u>  $-\frac{\text{Tr}(\Sigma_{k}^{(0)})}{\text{Tr}(\Sigma_{k}^{(0)})} \leq \varepsilon$  where  $\Sigma_{k}^{(0)}$  is the  $k \times k$  subset of  $\Sigma^{(0)}$  containing the largest diagonal values and  $\varepsilon$  is a set threshold; or, 380 <u>select k so that the kth eigenvector has a signal-to-noise ratio</u>  $SNR_k = \begin{cases} \frac{\sigma_k^{(0)}}{1 - \sigma_k^{(0)}} > -1, \text{ where } \sigma_k^{(0)} \text{ is the kth largest diagonal value of } \Sigma^{(0)}; \end{cases}$ 3: Form the  $n \times k$  matrix  $\Gamma^{*(0)} = \mathbf{S}_{A}^{1/2} \mathbf{W}_{k}^{(0)}$  where  $\mathbf{W}_{k}^{(0)}$  is a matrix of the first k columns of  $\mathbf{W}^{(0)}$  as ranked by the diagonal values of **Σ**<sup>(0)</sup>. 4: Perturb in the forward model the columns of  $\Gamma^{*(0)}$  to generate the  $m \times k$  reduced-dimension Jacobian matrix  $\mathbf{K}_{\alpha}^{(1)}$ ; 5: Form the  $m \times n$  reduced-rank Jacobian matrix  $\mathbf{K}_{\Pi}^{(1)} = \mathbf{K}_{\omega}^{(1)} \Gamma^{(0)}$ , where  $\Gamma^{(0)} = \mathbf{W}_{k}^{(0)}^{T} \mathbf{S}_{A}^{-1/2}$  and the corresponding  $n \times n$  reduced-rank Formatted: Font: Not Bold 385 averaging kernel matrix  $A_{\pi}^{(1)}$ ; <u>6: Let  $\mathbf{A}^{(0)} = \mathbf{A}_{\Pi}^{(1)}$  and repeat steps 1 to 5 to generate the final  $m \times n$  reduced-rank Jacobian matrix  $\mathbf{K}_{\Pi}^{(2)}$ .</u> 3 Results and discussion We demonstrate the reduced-dimension and reduced-rank Jacobian matrix construction methods in an analytical Bayesian 390 inversion of atmospheric methane columns observed by the GOSAT satellite over North America in July 2009. Although TROPOMI now provides higher density observations, using GOSAT allows us to use the inversion framework of Deleted: follow

Deleted: to

11

Maasakkers (2019). We construct a "native-resolution" inverse system at 1° x 1.25° grid cell resolution (n = 2098, top left

panel of Figure 1) against which we compare our reduced-dimension and reduced-rank methods. To demonstrate the applicability of the methods to higher-information observing systems such as TROPOMI, we artificially increase the information content of the GOSAT data by introducing an amplification factor  $\lambda > 1$  to the cost function that increases the weight of the observational terms:

$$\mathcal{J}_{\mathbf{x}}(\mathbf{x}) = (\mathbf{x} - \mathbf{x}_{A})^{\mathrm{T}} \mathbf{S}_{A}^{-1} (\mathbf{x} - \mathbf{x}_{A}) + \lambda (\mathbf{y} - \mathbf{F}(\mathbf{x}))^{\mathrm{T}} \mathbf{S}_{0}^{-1} (\mathbf{y} - \mathbf{F}(\mathbf{x})).$$

The amplification factor functionally decreases the observational error covariance, increasing the DOFS. We set  $\lambda = 5$ , which increases the native-resolution DOFS from <u>82 to 198</u>. Because of noise in the GOSAT data, this results in an overfit solution that is irrelevant in our demonstration.

We use the nested North American GEOS-Chem CTM version 12.4.0 as forward model to simulate atmospheric methane column concentrations at 1° x 1.25° resolution for July 2009. The 2098 1° x 1.25° grid cells constitute our native-resolution state vector. The model is driven with MERRA-2 meteorological fields (Bosilovich et al., 2016) from the NASA Global
Modeling and Assimilation Office. We use boundary conditions and initial conditions from a global posterior GEOS-Chem 4° x 5° simulation from Maasakkers et al. (2019). The GOSAT observations are from the University of Leicester version 7 CH4 proxy retrieval over land (Parker et al., 2011; Parker et al., 2015; ESA CCI GHG project team, 2018) for July 2009.

Prior emissions and observational error covariances are as described in Maasakkers et al. (2019). We assume uniform relative prior errors of 50%. The demonstration is sufficiently coarse-resolution and short that the native-resolution Jacobian matrix **K** can be explicitly computed with 2099 model runs. After constructing **K**, we use it as the forward model in lieu of additional GEOS-Chem simulations.

Figure 2 (top left panel) shows the native-resolution averaging kernel sensitivities, i.e., the diagonal elements of the native-resolution averaging kernel matrix A. As discussed in Section 2.3, the structure of the averaging kernel matrix is largely
determined by the prior error covariance matrix S<sub>A</sub> and by the observation density as reflected in both the observational covariance matrix S<sub>0</sub> and the Jacobian matrix K. This is apparent in the bottom panels of Figure 2, which show the distribution of the prior error standard deviations (left) and observation density (right). The absolute errors on the prior emissions estimate are largest for wetlands along the southeastern coast of the U.S. (Bloom et al., 2017). The variability in the observation density is driven by sampling frequency and retrieval success (Parker et al., 2015).

Figure 2 (top right panel) also shows the initial estimate of averaging kernel sensitivities of  $\mathbf{A}^{(0)}$  derived from the initial estimate of the Jacobian matrix  $\mathbf{K}^{(0)}$  constructed as described in Section 2.3. No forward model simulations were conducted to construct this initial estimate, yet the patterns of information content reproduce those given by the native-resolution

12

Deleted: J

ر13<sub>)</sub>

 Deleted: 40

 Deleted: 216

 Deleted: , which

| -( | Field Code Changed             |
|----|--------------------------------|
| -( | Deleted: Parker et al., 2011;  |
| -( | Deleted: and error covariances |

| (  | Deleted: defined |
|----|------------------|
| -( | Deleted: by      |
| (  | Deleted: by      |
| (  | Deleted: E       |

averaging kernel matrix **A** because of the strong dependence on the prior error standard deviations and the observation 440 density. **A** has a smoother structure than **A**<sup>(0)</sup> because of the effect of long-range transport in the CTM, <u>but this has little</u> effect on the leading patterns of information content.

- Figure 2. Averaging kernel sensitivities for the demonstration inversion of GOSAT observations with enhanced information content for July 2009. The top panels show the sensitivities given by the diagonal elements of the averaging kernel matrix A of the native-resolution inversion (left) and of the initial estimate of the averaging kernel matrix A<sup>(0)</sup> (right). The DOFS for each averaging kernel matrix are inset in the corresponding panel. The lower left panel shows the error standard deviations on the prior emissions estimate given by the square roots of the diagonal elements of S<sub>A</sub>. The lower right panel shows the GOSAT observation density.
- For this demonstration, we aim to reduce the number of forward model runs needed to construct the Jacobian matrix by at least a factor of four relative to the native-resolution inversion, from 2099 to ≈525 simulations. We first apply the reduced-dimension method to construct a Jacobian matrix on a multiscale grid generated with K-means clustering following Section 2.4. The resulting initial multiscale grid and reduced-dimension Jacobian matrix K<sup>1</sup><sub>RD</sub> constrain <u>380</u> clusters and required <u>381</u> model simulations. We disaggregate <u>J1</u> clusters with a sensitivity increase greater than 0.6 adding <u>65</u> native-resolution grid cells and model simulations. The resulting multiscale grid is shown in the top right panel of Figure 1. It has dimension <u>434</u> and the corresponding reduced-dimension Jacobian matrix K<sup>2</sup><sub>RD</sub> required <u>446</u> forward model simulations. <u>across 17</u>

| ( [ | Deleted: 4                                                                                     |
|-----|------------------------------------------------------------------------------------------------|
| (C  | Deleted: 530                                                                                   |
| C   | Deleted: s                                                                                     |
| C   | Deleted: 359                                                                                   |
| (   | Deleted: 470                                                                                   |
|     | Deleted: , where the excess simulations ensured the convergence f the DOFS. We disaggregate 16 |
| C   | Deleted: 16                                                                                    |
| (   | Deleted: 4                                                                                     |
| (   | Deleted: 64                                                                                    |
| C   | Deleted: 423                                                                                   |
| (   | Deleted: 534                                                                                   |

Deleted: which

We next apply the reduced-rank method (Section 2.5) to construct a reduced-rank approximation of the Jacobian matrix. We 490 calculate the dominant eigenvectors of the initial averaging kernel matrix estimate  $A^{(0)}$ , requiring that the signal-to-noise ratio of all eigenvectors be greater than 1.25. This yields  $k^{(0)} = 90$  eigenvectors, which account for 43% of the initial-

| 1  | Deleted: 2.5 |
|----|--------------|
| (  | Deleted: 74  |
| ~( | Deleted: 37  |

estimate DOFS. We perturb these eigenvectors in the forward model and construct the reduced-rank Jacobian matrix  $\mathbf{K}_{\Pi}^{(1)}$ . We then calculate the averaging kernel matrix  $\mathbf{A}_{\Pi}^{(1)}$  and its dominant eigenvectors, defining  $k^{(1)} = \underline{431}$  by requiring that the improved eigenvectors capture 97% of the information content defined by  $\mathbf{A}_{\Pi}^{(0)}$ . The resulting Jacobian matrix  $\mathbf{K}_{\Pi}^{(2)}$  has rank 510  $\approx \underline{431}$  and required 522 forward model simulations across two parallelized batches. We solve the inversion with  $\mathbf{K}_{\Pi}^{(2)}$  and find  $\underline{137}$  DOFS compared to the  $\underline{198}$  DOFS generated in the native-resolution inversion, achieving <u>69</u>% of the DOFS at a quarter of the computational cost.

The DOFS of the reduced-rank inversion are only moderately sensitive to the first<sub>e</sub> and second update thresholds, with a stronger dependence on the number of model runs conducted in the second update. Figure 4 shows the reduced-rank DOFS as a function of the number of first- and second-update forward model runs. Among all possible partitions of 522 total model runs (dashed line), our update scheme (starred) <u>nearly</u> maximizes the DOFS, <u>but the DOFS has only moderate sensitivity to the choice of partition</u>. Using a signal-to-noise ratio threshold of <u>0.75 or 1.75 instead of 1.25 (dots)</u>, decreases the reducedrank DOFS by <u>7%</u> Lowering the signal-to-noise ratio threshold increases the number of eigenvectors drawn from **A**<sup>(0)</sup>. which increases the effect of errors in the initial Jacobian matrix estimate **K**<sup>(0)</sup>. Increasing the threshold fails to exploit the information content of **A**<sup>(0)</sup>. More generally, applying a signal-to-noise ratio threshold of <u>1.25</u> in the first update maximizes the DOFS regardless of the number of model runs conducted in the second update. We show the DOFS generated by these optimal configurations as a function of the total number of forward model runs in the top panel of Figure 4. After only <u>300</u> simulations, the optimal reduced-rank inversion generates <u>101</u> DOFS, achieving half of the native-resolution DOFS at 14%

We solve the inversion (Eqs. (2) – (4)) using the reduced-rank Jacobian matrix K<sub>H</sub><sup>(2)</sup> and compare the posterior to the native-resolution solution. Figure 3 (right column) shows the distribution of the reduced-rank <u>posterior scaling factors</u> (top) and <u>averaging kernel sensitivities</u> (bottom) compared to the native-resolution inversion (left column). Because K<sub>H</sub><sup>(2)</sup> was
constructed on the basis of the dominant patterns of information content, it solves for the posterior scaling factors accurately in the areas of highest information content and defaults to the prior value (a scaling factor of one) elsewhere. As a result of the exclusion of grid cells with low native-resolution information content, the reduced-rank DOFS (137) are lower than native-resolution DOFS (198). However, in grid cells with large averaging kernel sensitivities, the reduced-rank inversion preserves most information content. <u>755</u> grid cells have reduced-rank averaging kernel sensitivities greater than 0.01 and generate <u>136</u> DOFS, <u>amounting to 83%</u> of the <u>163</u> DOFS generated by the <u>same</u> grid cells in the native-resolution inversion.

| Deleted: | 462                                     |
|----------|-----------------------------------------|
| Deleted: | 8.5                                     |
| Deleted: | 462                                     |
| Deleted: | 537                                     |
| Deleted: | 155                                     |
| Deleted: | 216                                     |
| Deleted: | 72                                      |
| Deleted: | -                                       |
| Deleted: | -                                       |
| Deleted: | 537                                     |
| Deleted: | 2                                       |
| Deleted: | 1                                       |
| Deleted: | or 4                                    |
| Deleted: | 2.5                                     |
| Deleted: | only 2-3                                |
| Deleted: | (from 155 to 150 and 152, respectively) |
| Deleted: | 2                                       |
| Deleted: | .5                                      |
| Deleted: | 275                                     |
| Deleted: | 108                                     |
| Deleted: | 3                                       |
| Deleted: | equations                               |
| Deleted: | averaging kernel sensitivities          |
| Deleted: | posterior scaling factors               |

| (                 | Deleted: 155 |
|-------------------|--------------|
| (                 | Deleted: 216 |
| (                 | Deleted: 699 |
| (                 | Deleted: 153 |
| -                 | Deleted: 7   |
| $\langle \rangle$ | Deleted: 175 |
| Ý                 | Deleted: se  |
|                   |              |

Figure 4. The sensitivity of the reduced-rank inversion DOFS to the number of forward model runs. The bottom panel shows the sensitivity of the DOFS to the partitioning of model runs between the first (x-axis) and second (y-axis) update. The lines represent the total number of simulations. Our inversion uses a signal-to-noise ratio of  $\frac{1}{2.25}$  for the first update and an information content threshold of  $\frac{97}{27}$ % for the second update (star), requiring  $\frac{522}{22}$  forward model runs and generating  $\frac{1}{37}$  DOFS, accounting for  $\frac{69}{6}$ % of the native-resolution DOFS at a quarter of the computational expense. Using a signal-to-noise ratio of  $\frac{0}{2.75}$  or 1.75 with the same total number of model runs for all optimal first- and second-update partitions.

| Deleted: 2.5    |  |
|-----------------|--|
| Deleted: 98.5   |  |
| Deleted: 537    |  |
| Deleted: 155    |  |
| Deleted: 72     |  |
| Deleted: 1 or 4 |  |

Figure 5. Comparison statistics between the reduced-rank and native-resolution inversions for the <u>erid cells with reduced-rank</u> averaging kernel sensitivity greater than 0.01. Individual panels compare binned counts for Jacobian matrix elements [ppb], posterior scaling factors [dimensionless], posterior error standard deviations [dimensionless], and averaging kernel sensitivities [bimensionless]. Correlation coefficients are inset and the 1:1 line (dotted) is shown.

Figure 5 shows a statistical comparison of the reduced-rank and native-resolution inversion results for grid cells with a reduced-rank averaging kernel sensitivity above 0.01. None of the reduced-rank quantities exhibit significant bias, as shown by comparison to the 1:1 line. The elements of the reduced-rank Jacobian matrix K<sub>h</sub><sup>(2)</sup> correspond closely with those of the native-resolution Jacobian matrix K (correlation coefficient R = 0.97). The strong correlation of the averaging kernel
sensitivities (R = 0.93) confirms that the reduced-rank inversion accurately identifies the native-resolution grid cells with the highest information content. The posterior errors and scaling factors agree well in these grid cells. The posterior error standard deviations correlate strongly (R = 0.94) due in part to the common contribution of the prior and observational error

| ( | Deleted: elements |                    |
|---|-------------------|--------------------|
|   |                   |                    |
|   |                   |                    |
|   |                   |                    |
|   |                   |                    |
|   |                   |                    |
|   |                   |                    |
|   |                   |                    |
|   |                   |                    |
| ( | Deleted: 96       |                    |
|   |                   |                    |
|   |                   |                    |
| 1 | Deleted: very     |                    |
| 1 | •                 | $ \longrightarrow$ |
|   | Deleted: 9        |                    |
|   |                   |                    |

covariance matrices (Eq. (3)). The outlier reduced-rank standard deviations tend to be larger than the native-resolution values, reflecting the error introduced by discarding information content. The posterior scaling factors also agree well but the correlation coefficient is smaller (R = 0.84) because of the smaller dynamical range and the propagation of errors from the
 posterior error covariance and Jacobian matrices (Eq. (2)). Negative scaling factors reflect the overfit from artificially increasing the information content but are of no consequence for our demonstration.

The reduced-dimension and reduced-rank methods reproduce the native-resolution inversion with a factor of <u>at least four</u> reduction in <u>total</u> computational cost. The reduced-dimension method generates lower DOFS but higher DOFS per state vector element due to the clustering of grid cells. The resulting posterior solution is exact on the multiscale grid and provides

better spatial coverage than the reduced-rank method\_at lower resolution. The reduced-rank method generates a higher-DOFS, higher-resolution approximation where the averaging kernel sensitivities are large. While the calculation of large Jacobian matrices can take advantage of parallel computing environments (Maasakkers et al., 2019), the iterative nature of both methods proposed here puts some limit on parallelization. The limit is greater for the reduced-dimension method, which
 requires an iteration for each cluster size added to the state vector. The reduced-rank method requires only two iterations. In both cases, these limitations may not be meaningful because the native-resolution Jacobian matrix is rarely generated in a

4 Conclusions

fully parallel environment in practice.

605

- We proposed two methods to conduct analytical high-resolution inversions of satellite observations of atmospheric 615 composition to infer emissions while maximizing information content and minimizing computational cost. The computational cost of analytical inversions is driven by the construction of the Jacobian matrix, which expresses the sensitivity of the observed concentrations to emissions. The Jacobian matrix is constructed numerically by conducting perturbation simulations with a chemical transport model (CTM) that serves as forward model for the inversion. Our methods exploit the dominant patterns of information content in the observing system to build the Jacobian matrix. The 620 reduced-dimension method constructs the Jacobian matrix on a multiscale grid that aggregates grid cells where information content is lowest. The reduced-rank method approximates the Jacobian matrix using the dominant patterns of information content, discarding the weaker patterns. Beyond the <u>atmospheric</u> application <u>presented here</u>, both methods can be applied more generally to the problem of efficient numerical approximation of high-dimension Jacobian matrices.
- Both methods use a two-step update to improve an initial, no-cost estimate of the Jacobian matrix and the corresponding averaging kernel matrix. The initial estimate of the Jacobian matrix is constructed<u>here</u> by assuming that observed atmospheric concentrations are most sensitive to local emissions. Because the averaging kernel matrix has a strong dependence on the prior error covariance <u>matrix</u> and observation density, this initial estimate can accurately quantify the fine

18

| Deleted: equation |  |
|-------------------|--|
|-------------------|--|

• Deleted: 9

Deleted: equation

Deleted: 4

Deleted: but Deleted: n Deleted: with higher DOFS and higher resolution

Deleted: present

structure of information content. The reduced-dimension method uses the initial estimate of the averaging kernel matrix to generate a multiscale grid that maintains native resolution where information content is highest and consolidates grid cells elsewhere. The forward model is used to build the Jacobian matrix on this grid, and the resulting reduced-dimension

averaging kernel matrix is compared to the initial estimate to identify the state vector elements where the forward model contributed the most information content. These elements are disaggregated to generate a second and final update. The reduced-rank method uses the initial estimate of the averaging kernel matrix to identify the dominant patterns of information content. These patterns are perturbed in the forward model, generating a first update of the Jacobian matrix. This update serves as the basis for a second and final update. In both methods, rapid convergence occurs after two updates.

We applied both methods in a demonstration inversion of GOSAT column methane observations over North America for July 2009 with artificially enhanced information content. We compared the results to a native-resolution inversion optimizing emissions on a 1° x 1.25° grid. Both methods successfully approximated the native-resolution results and decreased total computational cost by a factor of at least four, The reduced-dimension method produced only 50% of the native-resolution information content as measured by the degrees of freedom for signal (DOFS) due to spatial averaging but generated twice the DOFS per state vector element and avoided the correlated errors found in the native-resolution inversion. The reduced-rank method retained 70% of the native-resolution DOFS by solving the inversion accurately in the grid cells with the highest information content and defaulting to the prior emissions estimate elsewhere. In sensitivity tests, the reduced-rank method retained 50% of the native-resolution DOFS while decreasing computational cost by a factor of seven.

### 655 Code and data availability

All code and data are available at https://github.com/hannahnesser/reduced\_rank\_jacobian.

### Author contribution

HN and DJJ designed the study. HN performed the analysis. JDM provided all demonstration inversion data and supporting guidance. HN and MPS performed the simulations. HN, DJJ, JDM, TRS, MPS, YZ, and CHR discussed the methods and results. HN and DJJ prepared the manuscript with contributions from all co-authors.

### **Competing interests**

The authors declare that they have no conflict of interest.

| Deleteur | The forward model is applied to t |
|----------|-----------------------------------|
|          |                                   |
|          |                                   |
|          |                                   |
| Deleted: |                                   |
| Deletea: | and                               |
|          |                                   |
| Deleted: | 4                                 |
| Deleted: |                                   |
| Deleted: | 40                                |

Deleted:

### Acknowledgements

This work was funded by the NASA Carbon Monitoring System and by an NSF Graduate Fellowship to H\_<u>Nesser. Y. Zhang</u> is supported by NSFC Project 42007198 and Westlake University. We thank Daven Henze, Kevin Bowman, Michael Brenner, Cynthia Randles, Jeremy Brandman, and Laurent White for helpful discussions and Marc Bocquet for an insightful review. Deleted: 0

Deleted:

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
