# Peer review of "Reduced-Cost Construction of Jacobian Matrices for High-Resolution Inversions of Satellite Observations of Atmospheric Composition"

_Atmospheric Measurement Techniques, 2020_

## Referee Comment (RC1) · Marc Bocquet (Referee) · 15 Feb 2021

**1   Main remarks**

This is a well written paper. Nonetheless, there does not seem to be so much added value compared to previous publications on this topic. However, this is a nice illustration of this difficult set of ideas (reduction of the control space) and, which, to me, is very welcome and useful. Among possible improvements, I would list:

[Figure]

- Although it is rather fair as it is, the bibliography and references should be given more attention, be more complete and ultimately improved. Several key references that predate those cited in the manuscript, should be mentioned first.

- A couple of algorithms (in a proper algorithmic environment – typically a pseudo-code) could be provided for both methods described in the manuscript.

- Because this work's objective is the improvement in efficiency and to decrease the inversion's computational cost, the use of parallelism and modern computer architecture should be discussed.

Overall, I believe the manuscript only requires minor revisions but that they should be very carefully addressed.

**2  Suggestions and typos:**

1. l.14: "be orders of magnitude lower than its coverage suggests": Although I fully agree with the authors, the phrasing seems a bit excessive.

2. l.42: "The solution is generally obtained by minimizing a Bayesian cost function...": Rigorously speaking, there is no such thing as a "Bayesian cost function"; I would suggest: "The solution is generally obtained in a Bayesian framework by minimizing a cost function..." for instance.

3. l.46: "Methods of estimating the error exist (Bousserez and Henze, 2018; Evensen, 2009), but these approaches are computationally expensive, incomplete, and rarely applied in practice.": there are many earlier papers dealing with errors, including posterior errors, with objective estimation. For instance, among our own contributions: Koohkan and Bocquet (2012); Koohkan et al. (2013).

4. l.81-82: "Bocquet et al. (2011) defined a method to find the optimal multiscale grid from an array of all allowable grids, but this requires a large computational investment.": ok, agreed, but solutions had been proposed (and tested with success!) in the companion paper (Bocquet and Wu, 2011), see for instance Section 3.

5. l.84-87: "were subjective and did not consider the information content of the forward model or the observations. Reduced-rank methods (Bousserez and Henze, 2018; Spantini et al., 2015) generate an approximation of the posterior solution at the original dimension n by solving the inversion in the directions of highest information content. Spantini et al. (2015) assumed knowledge of the Jacobian matrix.": Absolutely but so does Bocquet et al. (2011), albeit in the physical space rather in a spectral space.

6. l.78-89: There are other key papers in reduction methods applied to source inverse modelling in atmospheric chemistry that should be mentioned. Those are based on reversible-jump MCMCs. I can think of Lunt et al. (2016); Liu et al. (2017).

7. l.91: "that minimize" ⟶ "that minimizes"

8. The introduction is concise but very well written. However, the references chosen in the introduction mostly refer to the authors' works. I am fine with additionally citing your own papers and recent/fresh contributions to the field, but you should at least cite the seminal or key papers for each main idea. For instance: l. 44-45: "This minimum is typically found using a numerical (variational) method, often employing the adjoint of the CTM to compute the cost function gradient (Henze et al., 2007)." : Citing Henze et al. (2007) is fine assuming you do not forget typically earlier works such as Elbern and Schmidt (2001); Quélo et al. (2005) and studies by Greg Carmichael et al.

9. l.101: "of Jacobian matrix construction." ⟶ "of the Jacobian matrix construction."

10. l.107: the notation (A lower index) for the prior is very confusing, since in the literature it very often points to the Analysis, i.e. the posterior. I understand that this is the one used by Clive Rodgers (Rodgers, 2000) or in the 1D retrieval community, but this is not the one used by the large majority. Moreover, it is also conflicting with the dedicated notation A for the averaging kernel (which does not refer to the prior but to the posterior). I would strongly suggest to change notation to make the manuscript easier and less confusing to read.

11. l.116: Same issue with $\mathbf{K}$ which is universally used as the Kalman gain matrix (Kalman, 1960), including in the geophysical inverse problems and data assimilation literature.

12. l.120; Equations (2, 3): you forgot the punctuation of the equations. Please check the whole manuscript and its equations.

13. l.172-175: "We will refer to the rate at which the information content accumulates as the number of eigenvectors increases as the information content spectrum.": More simply put, the spectrum is the ordered list of the eigenvalues.

14. l.229-231: I understood the point on clustering. Yet, it seems a bit vague to me. You could be more specific.

15. l.240-242: Again, this passage is not so clear and could be improved, although I guess I roughly understood.

16. Sections 2.4 and 2.5: I believe you could/should add a pseudo-code to each algorithm. The text is rather (though not entirely) clear and adding an algorithm would really help/reassure the reader. Obviously, these are the key sections of

the manuscript, so that it's worth investing time and (manuscript) space in such algorithms.

17. l.261: This was first proposed and proven in Bocquet et al. (2011), section 2.4.

18. l.337-344: The results for the reduced-dimension solution are somehow underwhelming; the resulting DOFS are quite low. Do you have a explanation for this? Or did I miss something?

19. You did not discuss at all the impact of time as you considered a static mesh for the emission. Can you discuss briefly the approximation that such assumption entails?

20. You did not discuss the patterns provided by the eigenvectors (main modes of the DOFS). Is it worth discussing this point?

21. l.425-426: You might want to have a look at solutions proposed in the meteorological data assimilation community to efficiently compute the Jacobian in high dimension, for instance Frolov et al. (2018).

22. I believe you should discuss parallelism of your algorithms and codes. Your paper is targeted at more efficient techniques – which will also depend on how well you are able to exploit parallelism. Please add a thorough discussion on the subject.

**References**

Bocquet, M., Wu, L., 2011. Bayesian design of control space for optimal assimilation of observations. II: Asymptotics solution. Q. J. R. Meteorol. Soc. 137, 1357–1368. doi:10.1002/qj.841.
Bocquet, M., Wu, L., Chevallier, F., 2011. Bayesian design of control space for optimal assimilation of observations. I: Consistent multiscale formalism. Q. J. R. Meteorol. Soc. 137, 1340–1356. doi:10.1002/qj.837.

Elbern, H., Schmidt, H., 2001. Ozone episode analysis by four-dimensional variational chemistry data assimilation. J. Geophys. Res. 106, 3569–3590.

Frolov, S., Allen, D.R., Bishop, C.H., Langland, R., Hoppel, K.W., Kuhl, D.D., 2018. First application of the local ensemble tangent linear model (LETLM) to a realistic model of the global atmosphere. Mon. Wea. Rev. 146, 2247–2270. doi:10.1175/MWR-D-17-0315.1.

Henze, D.K., Hakami, A., Seinfeld, J.H., 2007. Development of the adjoint of GEOS-Chem. Atmos. Chem. Phys. 7, 2413–2433. doi:10.5194/acp-7-2413-2007.

Koohkan, M.R., Bocquet, M., 2012. Accounting for representativeness errors in the inversion of atmospheric constituent emissions: Application to the retrieval of regional carbon monoxide fluxes. Tellus B 64, 19047. doi:10.3402/tellusb.v64i0.19047.

Koohkan, M.R., Bocquet, M., Roustan, Y., Kim, Y., Seigneur, C., 2013. Estimation of volatile organic compound emissions for Europe using data assimilation. Atmos. Chem. Phys. 13, 5887–5905. doi:10.5194/acp-13-5887-2013.

Liu, Y., Haussaire, J.M., Bocquet, M., Roustan, Y., Saunier, O., Mathieu, A., 2017. Uncertainty quantification of pollutant source retrieval: comparison of Bayesian methods with application to the Chernobyl and Fukushima-Daiichi accidental releases of radionuclides. Q. J. R. Meteorol. Soc. 143, 2886–2901. doi:10.1002/qj.3138.

Lunt, M.F., Rigby, M., Ganesan, A.L., Manning, A.J., 2016. Estimation of trace gas fluxes with objectively determined basis functions using reversible-jump Markov chain Monte Carlo. Geosci. Model Dev. 9, 3213–3229. URL: http://www.geosci-model-dev.net/9/3213/2016/, doi:10.5194/gmd-9-3213-2016.

Quélo, D., Mallet, V., Sportisse, B., 2005. Inverse modeling of $NO_x$ emissions at regional scale over northern France: Preliminary investigation of the second-order sensitivity. J. Geophys. Res. 110, D24310.

Rodgers, C.D., 2000. Inverse methods for atmospheric sounding. volume 2. World Scientific, Series on Atmospheric, Oceanic and Planetary Physics.

---

## Referee Comment (RC2) · Anonymous Referee #2 · 10 May 2021

Review of Reduced-Cost Construction of Jacobian Matrices for High Resolution Inversions of Satellite Observations of Atmospheric Composition.

My first comment (and very important one) here is that this paper has nothing to do with atmospheric measurement techniques and therefore its exact home is GMD and not AMT. Specifically, no "new" measurements are collected or discussed as part of this paper or for that matter no new measurement techniques are also suggested as part of this paper. [Outside the scope of the Journal]

The authors suggest two new techniques for reducing the cost of computing the Jacobian i.e., reduced rank and reduced dimension methods. First, these are not the only way to reduce the computational size of the problem.

(1) The generally accepted solution to reduce the size of the problem is the one suggested in the paper: "Measuring information content from observation for data assimilation: relative entropy versus Shannon entropy difference" and I would suggest the authors to review this paper. Thus, I would like to see the techniques suggested in this paper in comparison to those mentioned in the paper mentioned above. Note, these issues are nothing new and have been dealt with since 1974. (see paper the information content of remote measurements of atmospheric temperature by satellite infra-red radiometry and optimum radiometer configurations.). Eventually, it is the question of the information content of the observations and not reducing the size of the Jacobian or the information content as expressed through an Averaging Kernel. I would like to see the difference in the answer as received from the method described in Xu's paper in comparison to what is shown in this paper.

(2) Please also look at the paper "Stable Signal Recovery from Incomplete and Inaccurate Measurements from Candes, Romberg and Terence Tao" to understand the mathematical theory behind it. For application in atmospheric inversions see: A sparse reconstruction method for the estimation of multi-resolution emission fields via atmospheric inversion

(3) Following with the previous discussion if you have prior information, then you can aggregate grids where you do not have any chances of encountering methane fluxes without doing a two-step inversion. What is the point of solving for methane fluxes in the deserts of Nevada, Utah and Arizona (see Figure 2 in paper; you have regular grid) unless you expect deserts of Nevada to be big sources of methane emissions? For example, if you do this exercise globally then you would not be solving for methane fluxes in Sahara Desert (no unique information is provided by multitude of observations, even if theoretically a satellite can collect thousands of them). Hence even if the trace of

the averaging kernel might show that you can better resolve fluxes in the Sahara Desert solving for these fluxes would be just meaningless implying that you can aggregate your grid.

(4) Please also remember that once you go from coarser resolution to finer resolution your posterior variance of the inverse problem is guaranteed to increase. Hence, please explain or mathematically show how does the reduction in posterior variance translate from coarser resolution to finer resolution (not in terms of R i.e., correlation). Can an upper bound be found and does it have spatial structure i.e., what has happened to the error you obtained from the inversion (second part of equation 2)? Furthermore, what has happened to the trace of the averaging kernel. How has it distributed your trace at finer resolution?

---

## Author Comment (AC1)

**Responses to Reviewer 1**

We thank the reviewers for their comments and questions. Our responses are formatted as follows:

*The reviewer's comment/question (numbered) is written in black italic text.*

> Our responses are written in normal black text (indented).

> The revised text as it appears in the manuscript is written in normal blue text (indented), with relevant changes underlined.

Line numbers refer to the edited manuscript. We have also provided a tracked-changes document, but that has different line numbers.

*This is a well written paper. Nonetheless, there does not seem to be so much added value compared to previous publications on this topic. However, this is a nice illustration of this difficult set of ideas (reduction of the control space) and, which, to me, is very welcome and useful. Among possible improvements, I would list:*

- *Although it is rather fair as it is, the bibliography and references should be given more attention, be more complete and ultimately improved. Several key references that predate those cited in the manuscript, should be mentioned first.*
- *A couple of algorithms (in a proper algorithmic environment – typically a pseudocode) could be provided for both methods described in the manuscript.*
- *Because this work's objective is the improvement in efficiency and to decrease the inversion's computational cost, the use of parallelism and modern computer architecture should be discussed.*

*Overall, I believe the manuscript only requires minor revisions but that they should be very carefully addressed.*

Thank you for these comments. We responded to these points in the individual responses below.

*1. l.14: "be orders of magnitude lower than its coverage suggests": Although I fully agree with the authors, the phrasing seems a bit excessive.*

We changed "orders of magnitude" to "significantly."

The information content of satellite data may be much lower than its coverage would suggest…. (L15 – L16)

*2. l.42: "The solution is generally obtained by minimizing a Bayesian cost function...": Rigorously speaking, there is no such thing as a "Bayesian cost function"; I would suggest: "The solution is generally obtained in a Bayesian framework by minimizing a cost function..." for instance.*

Thanks for this suggestion; we've changed the text per your suggestion.

The solution is generally obtained in a Bayesian framework by minimizing a cost function…. (L44 – L45)

*3. l.46: "Methods of estimating the error exist (Bousserez and Henze, 2018; Evensen, 2009), but these approaches are computationally expensive, incomplete, and rarely applied in practice.": there are many earlier papers dealing with errors, including posterior errors, with objective estimation. For instance, among our own contributions: Koohkan and Bocquet (2012); Koohkan et al. (2013).*

Thanks for this comment. We added a reference to Chevallier et al. (2007), Meirink et al. (2008), and Koohkan et al. (2013). We did not find a description of posterior error estimation in Koohkan and Bocquet (2012).

Methods of estimating the error exist (e.g. Chevallier et al., 2007; Meirink et al., 2008; Koohkan et al., 2013), but these approaches are computationally expensive, incomplete, and rarely applied in practice. (L49 – L50)

4. l.81-82: "Bocquet et al. (2011) defined a method to find the optimal multiscale grid from an array of all allowable grids, but this requires a large computational investment.": ok, agreed, but solutions had been proposed (and tested with success!) in the companion paper (Bocquet and Wu, 2011), see for instance Section 3.

Thanks for your comment. We now cite *Bayesian Design of Control Space Part I* and *Part II* and more accurately describe the computational needs and optimality criteria of the methods proposed in those works.

Bocquet et al. (2011) and Bocquet and Wu (2011) defined a method to select a multiscale grid from a limited array of allowable grids that preserve resolution where the observations have the highest information content. (L87 – L89)

5. l.84-87: "were subjective and did not consider the information content of the forward model or the observations. Reduced-rank methods (Bousserez and Henze, 2018; Spantini et al., 2015) generate an approximation of the posterior solution at the original dimension n by solving the inversion in the directions of highest information content. Spantini et al. (2015) assumed knowledge of the Jacobian matrix.": Absolutely but so does Bocquet et al. (2011), albeit in the physical space rather in a spectral space.

See our response to comment 4.

6. l.78-89: There are other key papers in reduction methods applied to source inverse modelling in atmospheric chemistry that should be mentioned. Those are based on reversible-jump MCMCs. I can think of Lunt et al. (2016); Liu et al. (2017).

Thank you for your comment. We have added a citation to Rigby et al. (2011), Thompson and Stohl (2014), Ray et al. (2015), Lunt et al. (2016), and Liu et al. (2017).

Other approaches that decreased the dimension of the state vector assumed knowledge of the Jacobian matrix (e.g., Rigby et al., 2011; Thompson and Stohl, 2014; Ray et al., 2015; Lunt et al., 2016; Liu et al., 2017). (L91 – L93)

7. l.91: "that minimize" $-\rightarrow$ "that minimizes"

We write "Here we present two methods … that maximize the information content." We believe this is correct grammar, but we have modified the sentence for clarity.

Here we present two methods to construct the Jacobian matrix for a native $n$-dimensional state vector and maximize the information content of the inverse analysis using $k < n$ forward model simulations. (L99 – L100)

*8. The introduction is concise but very well written. However, the references chosen in the introduction mostly refer to the authors' works. I am fine with additionally citing your own papers and recent/fresh contributions to the field, but you should at least cite the seminal or key papers for each main idea. For instance: l. 44- 45: "This minimum is typically found using a numerical (variational) method, often employing the adjoint of the CTM to compute the cost function gradient (Henze et al., 2007)." : Citing Henze et al. (2007) is fine assuming you do not forget typically earlier works such as Elbern and Schmidt (2001); Quélo et al. (2005) and studies by Greg Carmichael et al.*

Thank you for this comment. We added a citation to Daescu et al. (2000), Elbern and Schmidt (2001), and Quélo et al. (2005) to the Henze et al. (2007) citation. We also added citations to other works as described in comments 3 and 6.

This minimum is typically found using a numerical (variational) method, often employing the adjoint of the CTM to compute the cost function gradient (e.g., Daescu et al., 2000; Elbern and Schmidt, 2001; Quélo et al., 2005; Henze et al., 2007). (L46 – L48)

*9. l.101: "of Jacobian matrix construction." $-\rightarrow$ "of the Jacobian matrix construction."*

Changed.

This section presents the reduced-dimension and reduced-rank methods of constructing the Jacobian matrix. (L109)

*10. l.107: the notation (A lower index) for the prior is very confusing, since in the literature it very often points to the Analysis, i.e. the posterior. I understand that this is the one used by Clive Rodgers (Rodgers, 2000) or in the 1D retrieval community, but this is not the one used by the large majority. Moreover, it is also conflicting with the dedicated notation A for the averaging kernel (which does not refer to the prior but to the posterior). I would strongly suggest to change notation to make the manuscript easier and less confusing to read.*

Thank you for this comment. While we recognize the benefits of the other notation commonly used in inversion discussions, we follow the notation of Rodgers (2000) and other analytical inversions that optimize methane emissions (i.e. Zhang et al. (2021), Maasakkers et al. (2021, 2019), Sheng et al. (2018), Fraser et al. (2014), etc.) for the sake of consistency and comparability.

*11. l.116: Same issue with K which is universally used as the Kalman gain matrix (Kalman, 1960), including in the geophysical inverse problems and data assimilation literature.*

See our response to comment 10.

*12. l.120; Equations (2, 3): you forgot the punctuation of the equations. Please check the whole manuscript and its equations.*

Thank you for catching this. We corrected this mistake by adding a comma to the end of equation (2) and a period to the end of equation (3). We also added commas to the end of equations (5), (7), (11), and (12).

*13. l.172-175: "We will refer to the rate at which the information content accumulates as the number of eigenvectors increases as the information content spectrum.": More simply put, the spectrum is the ordered list of the eigenvalues.*

Thanks for this suggestion. We've modified the language.

We will refer to the ordered list of the eigenvalues as the information content spectrum. (L181 – L182)

*14. l.229-231: I understood the point on clustering. Yet, it seems a bit vague to me. You could be more specific.*

Thanks for this comment. We've clarified the details of this method in an algorithm, per your suggestion in comment 16.

*15. l.240-242: Again, this passage is not so clear and could be improved, although I guess I roughly understood.*

See our response to comment 14.

*16. Sections 2.4 and 2.5: I believe you could/should add a pseudo-code to each algorithm. The text is rather (though not entirely) clear and adding an algorithm would really help/reassure the reader. Obviously, these are the key sections of the manuscript, so that it's worth investing time and (manuscript) space in such algorithms.*

Thank you for this suggestion. We've provided algorithms (below) describing the two methods (we didn't underline these additions for the sake of readability). We also added references to these algorithms to the text.

We apply this approach beginning with our initial estimate $\mathbf{K}^{(0)}$ (Section 2.3) in a two-step update that iteratively improves the multiscale grid. Algorithm 1 describes this process in detail. (L245 – L246)

**Algorithm 1: Reduced-dimension Jacobian matrix construction**
Given a native-resolution state vector with dimension $n$, a state vector encompassing the entire domain with dimension $n_{\mathrm{RD}} = 1$, and $\mathbf{A}^{(0)}$ and $\mathbf{A}_{\mathrm{RD}}^{(0)}$ the $n \times n$ and $1 \times 1$ initial estimates of the averaging kernel matrix, respectively, and an initial cluster size of one native-resolution grid cell:
1: Add the $J$ clusters with the highest diagonal values of $\mathbf{A}^{(0)}$ to the state vector and update its dimension $n_{\mathrm{RD}} = n_{\mathrm{RD}} + J$;

2: Perturb in the forward model those $J$ clusters and the background cluster to generate the $m \times n_{\text{RD}}$ reduced-dimension Jacobian matrix $\mathbf{K}_{\text{RD}}^{(1)}$ and the corresponding $n_{\text{RD}} \times n_{\text{RD}}$ averaging kernel matrix $\mathbf{A}_{\text{RD}}^{(1)}$;

3: If the difference in DOFS per cluster $\Delta\text{DPC} = \frac{\text{Tr}\left(\mathbf{A}_{\text{RD}}^{(1)}\right)}{n_{\text{RD}}} - \frac{\text{Tr}\left(\mathbf{A}_{\text{RD}}^{(0)}\right)}{n_{\text{RD}}-J} < \varepsilon$, where $\varepsilon$ is a set threshold, increase the cluster size (e.g., by aggregating together non-allocated, native-resolution grid cells using K-means clustering);

4: Let $\mathbf{A}_{\text{RD}}^{(0)} = \mathbf{A}_{\text{RD}}^{(1)}$ and repeat steps 1 to 4 until all native-resolution grid cells are allocated to the state vector;

5: Disaggregate the clusters with the largest increase in the diagonal values from $\mathbf{A}^{(0)}$ to $\mathbf{A}_{\text{RD}}^{(1)}$ as measured on the multiscale grid and update the state vector dimension $n_{\text{RD}}$;

6: Perturb in the forward model the disaggregated grid cells to generate the final $m \times n_{\text{RD}}$ reduced-dimension Jacobian matrix $\mathbf{K}_{\text{RD}}^{(2)}$.

(L255 – L267)

In an inverse system without a known Jacobian matrix, the reduced-rank Jacobian matrix approximation can be constructed in a two-step update that iteratively improves the patterns of information content used as perturbations. Algorithm 2 describes this process in detail. (L288 – L290)

**Algorithm 2: Reduced-rank Jacobian matrix construction**

Given a native-resolution state vector with dimension $n$, $\mathbf{A}^{(0)}$ an $n \times n$ initial estimate of the averaging kernel matrix, and $\mathbf{S}_A$ the $n \times n$ prior error covariance matrix:

1: Complete the eigendecomposition of the $n \times n$ matrix $\mathbf{Q}^{(0)} = \mathbf{S}_A^{-1/2}\mathbf{A}^{(0)}\mathbf{S}_A^{1/2} = \mathbf{W}^{(0)}\mathbf{\Sigma}^{(0)}\mathbf{W}^{(0)\text{T}}$;

2: Select $k$ so that $1 - \frac{\text{Tr}\left(\mathbf{\Sigma}_k^{(0)}\right)}{\text{Tr}\left(\mathbf{\Sigma}^{(0)}\right)} < \varepsilon$ where $\mathbf{\Sigma}_k^{(0)}$ is the $k \times k$ subset of $\mathbf{\Sigma}^{(0)}$ containing the largest diagonal values and $\varepsilon$ is a set threshold; or, select $k$ so that the $k$th eigenvector has a signal-to-noise ratio $\text{SNR}_k = \sqrt{\frac{\sigma_k^{(0)}}{1-\sigma_k^{(0)}}} > \sim 1$, where $\sigma_k^{(0)}$ is the $k$th largest diagonal value of $\mathbf{\Sigma}^{(0)}$;

3: Form the $n \times k$ matrix $\mathbf{\Gamma}^{*(0)} = \mathbf{S}_A^{1/2}\mathbf{W}_k^{(0)}$ where $\mathbf{W}_k^{(0)}$ is a matrix of the first $k$ columns of $\mathbf{W}^{(0)}$ as ranked by the diagonal values of $\mathbf{\Sigma}^{(0)}$;

4: Perturb in the forward model the columns of $\mathbf{\Gamma}^{*(0)}$ to generate the $m \times k$ reduced-dimension Jacobian matrix $\mathbf{K}_\omega^{(1)}$;

5: Form the $m \times n$ reduced-rank Jacobian matrix $\mathbf{K}_\Pi^{(1)} = \mathbf{K}_\omega^{(1)}\mathbf{\Gamma}^{(0)}$, where $\mathbf{\Gamma}^{(0)} = \mathbf{W}_k^{(0)\text{T}}\mathbf{S}_A^{-1/2}$, and the corresponding $n \times n$ reduced-rank averaging kernel matrix $\mathbf{A}_\Pi^{(1)}$;

6: Let $\mathbf{A}^{(0)} = \mathbf{A}_\Pi^{(1)}$ and repeat steps 1 to 5 to generate the final $m \times n$ reduced-rank Jacobian matrix $\mathbf{K}_\Pi^{(2)}$.

(L305 – L316)

*17. l.261: This was first proposed and proven in Bocquet et al. (2011), section 2.4.*

Thank you for this comment. We added a reference to Bocquet et al. (2011).

Bousserez and Henze (2018), following Bocquet et al. (2011), show that the reduced-dimension Jacobian matrix $\mathbf{K}_\omega$ is given by $\mathbf{K}_\omega = \mathbf{K}\boldsymbol{\Gamma}^*$ and the reduced-rank Jacobian matrix $\mathbf{K}_\Pi$ by $\mathbf{K}_\Pi = \mathbf{K}\boldsymbol{\Pi} = \mathbf{K}\boldsymbol{\Gamma}^*\boldsymbol{\Gamma}$. (L284 – L285)

*18. l.337-344: The results for the reduced-dimension solution are somehow underwhelming; the resulting DOFS are quite low. Do you have a explanation for this? Or did I miss something?*

Thank you for this question. We found and fixed a problem with how we calculated the error covariance at reduced-dimension, which increased the DOFS slightly. We've also clarified the dependence of DOFS on the dimension of the state vector.

The reduced-dimension solution generates fewer DOFS (95) than the native-resolution solution (198) because the DOFS depend on the dimension of the state vector. When comparing the DOFS per cluster, a dimension-independent measure, the reduced-dimension solution generates more than twice the value of the native-resolution solution (0.22 compared to 0.09), reflecting the consolidation of information content. (L383 – L386)

*19. You did not discuss at all the impact of time as you considered a static mesh for the emission. Can you discuss briefly the approximation that such assumption entails?*

Thanks for asking. We clarified that these methods can be applied to temporally-resolved state vectors in the introduction to the methods section.

For the purposes of illustration, we take the state vector to be a gridded field of static emissions, but the methods apply to temporally variable emissions and more generally to any state vector. (L113 – L114)

*20. You did not discuss the patterns provided by the eigenvectors (main modes of the DOFS). Is it worth discussing this point?*

Thanks for asking. As you note, the eigenvectors are the main modes of the DOFS. As a result, we find that the leading eigenvectors show little information that is not expressed in the averaging kernel sensitivities. As a result, we discuss the averaging kernel sensitivities directly rather than providing a separate discussion of the patterns.

*21. l.425-426: You might want to have a look at solutions proposed in the meteorological data assimilation community to efficiently compute the Jacobian in high dimension, for instance Frolov et al. (2018).*

Thank you for this recommendation. We looked through the work of Frolov and Bishop and believe that the ensemble-based approaches are not applicable to the case considered here, which assumes the use of a deterministic chemical transport model.

*22. I believe you should discuss parallelism of your algorithms and codes. Your paper is targeted at more efficient techniques – which will also depend on how well you are able to exploit parallelism. Please add a thorough discussion on the subject.*

Thank you for this comment. We added statements detailing the parallelization used in each method and comparing the parallelization potential of each method.

[The resulting multiscale grid] has dimension 434 and the corresponding reduced-dimension Jacobian matrix $\mathbf{K}_{RD}^{(2)}$ required 446 forward model simulations across 17 parallelized batches.... (L371 – L373)

The resulting Jacobian matrix $\mathbf{K}_{\Pi}^{(2)}$ has rank $\approx$431 and required 522 forward model simulations across two parallelized batches. (L397 – L398)

The reduced-dimension and reduced-rank methods reproduce the native-resolution inversion with a factor of at least four reduction in total computational cost. The reduced-dimension method generates lower DOFS but higher DOFS per state vector element due to the clustering of grid cells. The resulting posterior solution is exact on the multiscale grid and provides better spatial coverage than the reduced-rank method at lower resolution. The reduced-rank method generates a higher-DOFS, higher-resolution approximation where the averaging kernel sensitivities are large. While the calculation of large Jacobian matrices can take advantage of parallel computing environments (Maasakkers et al., 2019), the iterative nature of both methods proposed here puts some limit on parallelization. The limit is greater for the reduced-dimension method, which requires an iteration for each cluster size added to the state vector. The reduced-rank method requires only two iterations. In both cases, these limitations may not be meaningful because the native-resolution Jacobian matrix is rarely generated in a fully parallel environment in practice. (L453 – L462)

---

## Author Comment (AC2)

**Responses to Reviewer 2**

We thank the reviewer for their comments and questions. Our responses are formatted as follows:

*The reviewer's comment/question (numbered) is written in black italic text.*

> Our responses are written in normal black text (indented).

> The revised text as it appears in the manuscript is written in normal blue text (indented), with relevant changes underlined.

Line numbers refer to the edited manuscript. We have also provided a tracked-changes document, but that has different line numbers.

*My first comment (and very important one) here is that this paper has nothing to do with atmospheric measurement techniques and therefore its exact home is GMD and not AMT. Specifically, no "new" measurements are collected or discussed as part of this paper or for that matter no new measurement techniques are also suggested as part of this paper. [Outside the scope of the Journal]*

> Thank you for your comment. Our paper concerns the interpretation of observations to infer secondary quantities (e.g., emissions). Other similar papers have been published by AMT in the past (e.g., Varon et al. 2018, Alden et al. 2018).

*The authors suggest two new techniques for reducing the cost of computing the Jacobian i.e., reduced rank and reduced dimension methods. First, these are not the only way to reduce the computational size of the problem.*

*(1) The generally accepted solution to reduce the size of the problem is the one suggested in the paper: "Measuring information content from observation for data assimilation: relative entropy versus Shannon entropy difference" and I would suggest the authors to review this paper. Thus, I would like to see the techniques suggested in this paper in comparison to those mentioned in the paper mentioned above. Note, these issues are nothing new and have been dealt with since 1974. (see paper the information content of remote measurements of atmospheric temperature by satellite infra-red radiometry and optimum radiometer configurations.). Eventually, it is the question of the information content of the observations and not reducing the size of the Jacobian or the information content as expressed through an Averaging Kernel. I would like to see the difference in the answer as received from the method described in Xu's paper in comparison to what is shown in this paper.*

> Thank you for your suggestion. Xu (2007) describes the dependence of two measures of information content (the Shannon and relative entropy differences) on optimal reductions in the dimension of the observation vector. We clarified the dependence of the computational cost on the dimension of the state vector. We also added references to Xu et al. (2007) to the introduction and to our discussion of measures of information content.

> When $m \gg n$, as for inversions of satellite observations, the Jacobian can be constructed column-wise by conducting $n + 1$ CTM simulations to perturb each of the state vector elements $x_i$ and obtain the corresponding column $\partial \mathbf{y}/\partial x_i$. (L58 – L60)

> Several methods have been proposed to decrease the computational cost of high-resolution analytical inversions by optimally reducing the dimension or rank of the observations or state vector. Approaches that reduce the dimension of the observation vector (e.g., Xu, 2007) reduce the computational cost of solving the inversion but not of constructing the Jacobian matrix. Approaches that decrease the dimension of the state vector lower the cost of both computations. (L82 – L85)

> The fraction of information content explained by the first $i$ columns of $\mathbf{\Gamma}^*$ is the sum of the $i$ largest eigenvalues divided by the total DOFS (Bousserez and Henze, 2018). The

eigenvalues can also be related to other measures of information content, including the Shannon and relative entropy differences (Rodgers, 2000; Xu, 2007). (L178 – L181)

*(2) Please also look at the paper "Stable Signal Recovery from Incomplete and Inaccurate Measurements from Candes, Romberg and Terence Tao" to understand the mathematical theory behind it. For application in atmospheric inversions see: A sparse reconstruction method for the estimation of multi-resolution emission fields via atmospheric inversion*

Thanks for your comment. We added a citation to Ray et al. (2015) (*A sparse reconstruction method*).

Other approaches that decreased the dimension of the state vector assumed knowledge of the Jacobian matrix (e.g., Rigby et al., 2011; Thompson and Stohl, 2014; Ray et al., 2015; Lunt et al., 2016; Liu et al., 2017). (L91 – L93)

*(3) Following with the previous discussion if you have prior information, then you can aggregate grids where you do not have any chances of encountering methane fluxes without doing a two-step inversion. What is the point of solving for methane fluxes in the deserts of Nevada, Utah and Arizona (see Figure 2 in paper; you have regular grid) unless you expect deserts of Nevada to be big sources of methane emissions? For example, if you do this exercise globally then you would not be solving for methane fluxes in Sahara Desert (no unique information is provided by multitude of observations, even if theoretically a satellite can collect thousands of them). Hence even if the trace of the averaging kernel might show that you can better resolve fluxes in the Sahara Desert solving for these fluxes would be just meaningless implying that you can aggregate your grid.*

Thank you for your comment. We have clarified that the averaging kernel sensitivities are low in areas known to have low emissions. The reduced-dimension method therefore functionally considers both the distribution of prior emissions and the observational density to generate a multiscale grid.

$\mathbf{A}$ can be calculated as $\mathbf{A} = \mathbf{I} - \hat{\mathbf{S}}\mathbf{S}_A^{-1}$ or equivalently as

$$\mathbf{A} = \mathbf{S}_A \mathbf{K}^T (\mathbf{K}\mathbf{S}_A\mathbf{K}^T + \mathbf{S}_O)^{-1}\mathbf{K}. \tag{4}$$

Equation (4) expresses the dependence of the averaging kernel matrix on the forward model and both error covariance matrices. The diagonal elements of $\mathbf{A}$ are commonly referred to as the averaging kernel sensitivities. They are highest in highly observed locations with uncertain, high emissions and lowest in poorly observed areas or in regions known to have low emissions. (L131 – L139)

*(4) Please also remember that once you go from coarser resolution to finer resolution your posterior variance of the inverse problem is guaranteed to increase. Hence, please explain or mathematically show how does the reduction in posterior variance translate from coarser resolution to finer resolution (not in terms of R i.e., correlation). Can an upper bound be found and does it have spatial structure i.e., what has happened to the error you obtained from the*

*inversion (second part of equation 2)? Furthermore, what has happened to the trace of the averaging kernel. How has it distributed your trace at finer resolution?*

Thank you for your comment. We aren't sure what you mean to ask here because we go from finer resolution to coarser resolution, not vice versa. We believe your question may be answered by the bottom row of Figure 3, which shows the distribution of the trace of the averaging kernel (the averaging kernel sensitivities) in each of the proposed methods. We believe that this may also answer your question about the distribution of the posterior error, since the averaging kernel is a measure of the relative reduction in error from the prior to the posterior: $\mathbf{A} = \mathbf{I} - \hat{\mathbf{S}}\mathbf{S}_A^{-1}$.